# A Closer Look at Generalized BH Algorithm for Out-of-Distribution Detection

**Xinsong Ma** [1]   **Jie Wu** [1]   **Weiwei Liu** [1]

## Abstract

Out-of-distribution (OOD) detection is a crucial task in reliable and safety-critical applications. Previous studies primarily focus on developing score functions while neglecting the design of decision rules based on these scores. A recent work (Ma et al., 2024) is the first to highlight this issue and proposes the generalized BH (g-BH) algorithm to address it. The g-BH algorithm relies on empirical p-values, with the calibrated set playing a central role in their computation. However, the impact of calibrated set on the performance of g-BH algorithm has not been thoroughly investigated. This paper aims to uncover the underlying mechanisms between them. Theoretically, we demonstrate that conditional expectation of true positive rate (TPR) on calibrated set for the g-BH algorithm follows a beta distribution, which depends on the prescribed level and size of calibrated set. This indicates that a small calibrated set tends to degrade the performance of g-BH algorithm. To address the limitation of g-BH algorithm on small calibrated set, we propose a novel ensemble g-BH (eg-BH) algorithm which integrates various empirical p-values for making decisions. Finally, extensive experimental results validate the effectiveness of our theoretical findings and demonstrate the superiority of our method over g-BH algorithm on small calibrated set.

## 1. Introduction

Out-of-Distribution (OOD) detection is a critical task in machine learning and computer vision (Hendrycks & Gimpel, 2017; Liu et al., 2020). It addresses the challenge of determining whether a given input sample belongs to the same distribution as the training data (Hendrycks et al., 2022; Djurisic et al., 2023). In real-world applications, models trained on in-distribution (ID) data from a specific domain often encounter OOD data from unseen distributions during deployment. This discrepancy between the training and deployment leads to poor model performance (Liang et al., 2018; Sastry & Oore, 2020; Kaur et al., 2022). The importance of OOD detection has grown with the increasing reliance on deep learning models in safety-critical applications, such as autonomous driving (Li et al., 2022) and medical diagnosis (Frolova et al., 2022).

Numerous studies have proposed various methods to address the OOD detection problem (Liu et al., 2020; 2023; Regmi et al., 2024; Lu et al., 2024). These methods mainly focus on designing the score functions which enables to learn critical discriminative information in training data. A recent work (Ma et al., 2024) is the first to point out that existing OOD detection methods neglect the systematic study of the decision rule based on the score functions, and propose a novel generalized BH (g-BH) algorithm to tackle this problem. The g-BH algorithm establishes a connection between score functions and multiple hypothesis testing framework through the empirical p-values. The calibrated set is crucial for computing empirical p-values and thus has a profound impact on the detection performance of the g-BH algorithm. However, to the best of our knowledge, the impact of the calibrated set on the performance of the g-BH algorithm remains unexplored. This paper aims to address the above issue.

Intuitively, a larger calibration set enhances the performance of the g-BH algorithm. Our experimental results in Figure 1 confirm this conjecture: as the size of calibrated set increases, both the TPR and F1-score monotonically increase. Theoretically, we demonstrate that the TPR expectation conditional on calibrated set for the g-BH algorithm follows a beta distribution, with its shape parameters determined by the prescribed significance level and the size of calibrated set. This shows that a smaller calibrated set tends to degrade the detection performance of the g-BH algorithm. To address the limitation of the g-BH algorithm on small calibrated set, we propose a novel ensemble g-BH (eg-BH) algorithm which integrates multiple empirical p-values for decision-making. Moreover, we extend the theoretical results on the g-BH algorithm from (Ma et al., 2024) and

[1]School of Computer Science, National Engineering Research Center for Multimedia Software, Institute of Artificial Intelligence and Hubei Key Laboratory of Multimedia and Network Communication Engineering, Wuhan University, Wuhan, China. Correspondence to: Weiwei Liu <liuweiwei863@gmail.com>.

*Proceedings of the 42nd International Conference on Machine Learning*, Vancouver, Canada. PMLR 267, 2025. Copyright 2025 by the author(s).

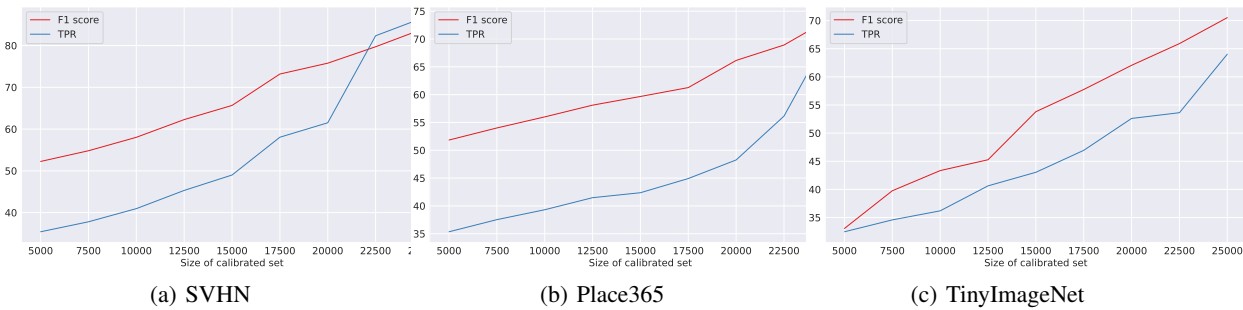

*Figure 1.* OOD detection performance of the g-BH algorithm with varying size of calibrated set. The score function is the Energy (Liu et al., 2020). The x-axis corresponds to the size of the calibrated set, and the y-axis represents the values of metrics.

demonstrate that the eg-BH algorithm controls the false discovery rate (FDR) for p-values without clear structural dependence.

Finally, we conduct extensive experiments to verify the theoretical results on the conditional expectation of TPR and the effectiveness of the eg-BH algorithm. Experimental results demonstrate the superiority of our method over the g-BH algorithm on small calibrated set.

We summarize our core contributions as follows:

- We empirically find that a larger calibrated set improves the performance of the g-BH algorithm, whereas a smaller calibrated set adversely affects its performance.

- We theoretically demonstrate that the TPR expectation conditional on the calibrated set follows a beta distribution, with its shape parameters determined by the prescribed significance level and the size of the calibrated set, which supports our findings.

- To address the limitation of the g-BH algorithm on the small calibrated set, we propose a novel eg-BH algorithm that integrates multiple empirical p-values for decision-making. Besides, our theoretical results provide a statistical guarantee for the integrated p-values in our eg-BH algorithm.

- Extensive experimental results demonstrate the superiority of our method over the g-BH algorithm on small calibrated set.

## 2. Background

We donote by $\mathcal{X} \subseteq \mathbb{R}^d$ the feature space and $\mathcal{Y} = \{1, 2, 3, \ldots, K\}$ the label space with unknown joint distribution $\mathcal{P}$, and $\mathcal{X}$ has marginal distribution $\mathcal{D}_x$.

During the prediction phase, it is typically assumed that the testing data are drawn from the same distribution $\mathcal{D}_x$ as the training data. However, in practical applications, test inputs may originate from unseen distributions, where the corresponding label space may be disjoint from $\mathcal{Y}$. These OOD samples should be identified and excluded from prediction.

The objective of OOD detection is to identify OOD examples in the testing set. In prior work, the OOD detection task is formulated as a binary decision problem:

$$\phi(x) = \begin{cases} ID, & \text{if } s(x) \geq s^* \\ OOD, & \text{if } s(x) < s^* \end{cases} \quad (1)$$

where $s(\cdot)$ is the score function and the threshold $s^*$ is empirically selected so that the ture positive rate (TPR) on ID validation set is $95\%$ before testing (Sun et al., 2022; Wei et al., 2022).

Prior work on OOD detection has primarily focused on designing powerful score functions to capture discriminative information in ID data (Hendrycks & Gimpel, 2017; Liu et al., 2020; Djurisic et al., 2023; Liu et al., 2023). However, Ma et al. (2024) highlight that these studies lacks systematic research on the decision rule on the score functions. Moreover, the decision rule in Eq (1) is empirical and lacks theoretical guarantee for its outputs. Different from the previous studies, Ma et al. (2024) studies the OOD detection problem from the perspective of multiple hypothesis testing, and propose the g-BH algorithm to tackle it.

## 3. Multiple Hypothesis Testing Framework for OOD Detection

We first introduce the hypothesis testing framework for OOD detection in Ma et al. (2024). For mathematical convenience, we follow the notations of Ma et al. (2024). Given a testing set $\mathcal{T}^{test} = \{X_1^{test}, X_2^{test}, \ldots, X_n^{test}\}$, For $i = 1, \cdots, n$, the OOD detection task is formulated as the following multiple hypothesis testing problem:

$$H_{1;0} : X_1^{test} \sim \mathcal{D}_x, \quad H_{1;1} : X_1^{test} \nsim \mathcal{D}_x$$
$$\cdots \cdots \quad (2)$$
$$H_{n;0} : X_n^{test} \sim \mathcal{D}_x, \quad H_{n;1} : X_n^{test} \nsim \mathcal{D}_x$$

where $H_{i;0}$ and $H_{i;1}$ are called null hypothesis and alternative hypothesis, respectively. Then, if $H_{i;0}$ is rejected, we declare that $X_i^{test}$ is OOD.

In statistics, the decision to accept or reject the null hypothesis is made based on the concept of the *p-value*, which is generally defined as follows:

**Definition 3.1. [p-value (Casella & Berger, 2002)]** Given a sample $\widetilde{X}$ [4]. A statistic $p(\widetilde{X})$ is called p-value corresponding to the null hypothesis $H_0$, if $p(\widetilde{X})$ satisfies

$$\mathbb{P}[p(\widetilde{X}) \leq t|H_0] \leq t \tag{3}$$

for every $0 \leq t \leq 1$.

If the statistic $p(\widetilde{X})$ follows the uniform distribution on $(0, 1)$ under the null hypothesis, it is a valid p-value. A small p-value typically provides strong evidence against the null hypothesis. It is noteworthy that the p-value has clear statistical interpretation. For example, if the p-value of a OOD testing example $X_i^{test}$ is 0.01, this implies that, for any subsequent testing example $X_j^{test}$, the probability that $X_j^{test}$ is more similar to OOD data than $X_i^{test}$ is 0.01. In other words, it is highly unlikely to find an example less similar to the OOD data than $X_i^{test}$. Hence, it provides strong evidence that $X_i^{test}$ is OOD.

*Remark* 3.2. In statistics, the following terminology characterizes the distribution of null p-values: if $\mathcal{P}[p(\widetilde{X}) \leq t|H_0] = t$, the p-value $p(\widetilde{X})$ is called exact or uniform; if $\mathcal{P}[p(\widetilde{X}) \leq t|H_0] < t$, $p(\widetilde{X})$ is called conservative. Compared to an exact p-value, a conservative one tends to understate the evidence against the null Hypothesis.

Based on the work (Benjamini & Hochberg, 1995), Ma et al. (2024) propose the g-BH algorithm to tackle the OOD detection problem. Define two function classes:

$$\mathcal{F}_1 = \{f(x) : f_+(0) = 0, f'(x) > 0, \int_0^1 \frac{1}{f(x)} \, dx \leq 1\}$$

$$\mathcal{F}_2 = \{f(x) : f_+(0) = 0, f'(x) \geq 1\},$$

where $f_+(0) = \lim_{x \to 0+} f(x)$ for $x \in (0, 1)$. Based on $\mathcal{F}_1$ and $\mathcal{F}_2$, the g-BH algorithm is defined as follows:

**Definition 3.3 (g-BH algorithm (Ma et al., 2024)).** Given the p-values $p_1, p_2, \cdots, p_n$ corresponding to the null hypotheses $H_{1;0}, H_{2;0}, \cdots, H_{n;0}$, let $p_{(i)}$ be the $i$-th order statistics from the smallest to the largest. For a pre-specified level $\alpha \in (0, 1)$, define

$$i_{g-BH}^* = \max\{i \in [n] : f(p_{(i)}) \leq \frac{i}{n}\alpha\}, \tag{4}$$

where $f(\cdot) \in \mathcal{F}_1 \cup \mathcal{F}_2$. Then, the null hypothesis $H_{(i);0}$ is rejected if $i \leq i_{g-BH}^*$.

---

[4] A sample means a sequence of examples.

Ma et al. (2024) demonstrate that if p-values are independent or positive regression dependence on subset (PRDS), the g-BH algorithm controls the FDR at prescribed level $\alpha$. FDR is defined as

$$\text{FDR} = \mathbb{E}\left[\frac{|\mathcal{R} \cap \mathcal{H}_0|}{\max\{1, |\mathcal{R}|\}}\right]$$

where $\mathcal{R}$ is the set of indices of the rejected null hypotheses and $\mathcal{H}_0$ is the set of indices for the true null hypotheses. (Ma et al., 2024) demonstrates that the g-BH algorithm can control the FDR at a prescribed level if p-values are mutually independent or satisfy the positive regression dependence on subset (PRDS) condition (Benjamini & Yekutieli, 2001).

## 4. Impact of Calibrated Set on Generalized BH Algorithm

In most multiple hypothesis testing literature (Benjamini & Hochberg, 1995; Benjamini & Yekutieli, 2001; Blanchard & Roquain, 2008; Delattre & Roquain, 2015; Cao et al., 2022), the p-values or the distribution of the testing statistic are assumed to be known. Denote by $F(\cdot)$ the cumulative distribution function of $s(X)$ where $s(\cdot)$ is the score function and $X \sim \mathcal{D}_x$. Then, for a given example $X^{test}$, its p-value can be expressed as

$$p(X^{test}) = \mathbb{P}_{X \sim \mathcal{D}_x}(s(X) \leq s(X^{test}))$$
$$= F(s(X^{test})). \tag{5}$$

According to the Definition 2, under the $H_0$ ($X^{test}$ is the ID data), we have

$$\mathbb{P}\left(F(s(X^{test})) \leq x\right) = \mathbb{P}\left(s(X^{test}) \leq F^{-1}(x)\right)$$
$$= F(F^{-1}(x)) = x,$$

where $F^{-1}(\cdot)$ is the inverse function of $F(\cdot)$. Therefore, the random variable $F(s(X^{test}))$ follows the uniform distribution on $(0, 1)$, namely, $p(X^{test})$ is a valid p-value and is exact. Obviously, small score $s(X^{test})$ results in a small p-value, which aligns with the classical setting of OOD detection in Eq.(1) and the interpretation of the p-value.

However, in the context of the OOD detection, we often have little prior information about underlying distribution $F(\cdot)$. Hence, Ma et al. (2024) propose using the empirical p-values in the g-BH algorithm, which is a nonparametric estimation method for the p-value $p(X^{test})$. Given a calibrated set $\mathcal{T}^{cal} = \{X_1^{cal}, X_2^{cal}, \ldots, X_m^{cal}\}$ consisting of the ID examples, for a testing example $X_i^{test}$, the empirical p-value $p_i$ corresponding to null hypothesis $H_{i;0}$ is defined as

$$p_i = \hat{p}(X_i^{test}) = \frac{\sum_{j=1}^m \mathbb{1}(s(X_j^{cal}) \leq s(X_i^{test})) + 1}{m + 1}, \tag{6}$$

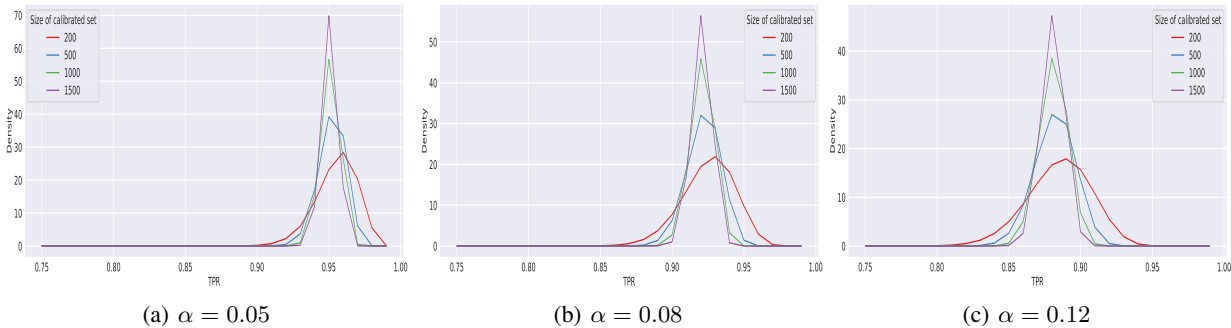

$$\text{(a) } \alpha = 0.05 \qquad\qquad \text{(b) } \alpha = 0.08 \qquad\qquad \text{(c) } \alpha = 0.12$$

*Figure 2.* Density distribution functions of the TPR conditional on calibrated set for the g-BH algorithm with varying level $\alpha$ and size $m$ of calibrated set.

where $s(\cdot)$ is a certain score function. According to Arlot et al. (2010), we can easily verify that empirical p-value in Eq. (6) satisfies the Definition 3.1.

Note that the g-BH algorithm directly makes the decisions based on the empirical p-values. Therefore, the calibrated set plays a critical role in the OOD detection performance of g-BH algorithm. However, to the best of our knowledge, there is no literature that thoroughly investigates the influence of calibrated set on the performance of the g-BH algorithm. This paper aims to systematically study this important problem.

In this paper, we focus on the situation where only ID data is available before testing. Thus, we first investigate how the size $m$ of calibrated set influences the conditional expectation of TPR on the calibrated set $\mathcal{T}^{cal}$ for the g-BH algorithm: $\mathbb{E}(\text{TPR}|\mathcal{T}^{cal})$.

To derive the distribution characteristics of conditional expectation of TPR, we need the concept of *empirical distribution function*. Given the calibrated set $\mathcal{T}^{cal} = \{X_1^{cal}, X_2^{cal}, \ldots, X_m^{cal}\}$, for any input $x$, the empirical distribution $\hat{F}(\cdot)$ of score function $s(\cdot)$ on $\mathcal{T}^{cal}$ can be expressed as

$$\hat{F}(x) = \frac{1}{m}\sum_{i=1}^{n} \mathbb{1}(s(X_i^{cal}) \leq s(x))$$

$$= \begin{cases} 0, & \text{if } s(x) < s(X_{(1)}^{cal}) \\ \frac{k}{m} & \text{if } s(X_{(k)}^{cal}) \leq s(x) < s(X_{(k+1)}^{cal}) \\ 1, & \text{if } s(x) \geq s(X_{(m)}^{cal}), \end{cases}$$

where $k = 1, 2, \cdots, m$ and $X_{(k)}^{cal}$ is the $k$-th order statistic of $X_1^{cal}, X_2^{cal}, \ldots, X_m^{cal}$ from the smallest to the largest. Obviously, we have

$$\hat{F}(X_{(k)}^{cal}) = \frac{k}{n}.$$

In addition, we denote $[\cdot]$ the floor function. Then, we have the following theoretical result.

**Theorem 4.1.** *Given the calibrated set $\mathcal{T}^{cal} = \{X_1^{cal}, X_2^{cal}, \ldots, X_m^{cal}\}$ and the score function $s(\cdot)$ that is continuous. For a prescribed level $\alpha$, the density function of $\mathbb{E}(\text{TPR}|\mathcal{T}^{cal})$ for the g-BH algorithm can be expressed as*

$$f_{tpr}(x) = \begin{cases} m\binom{m-1}{\beta-1}x^{m-\beta}(1-x)^{\beta-1} & \text{if } 0 < x < 1 \\ 0 & \text{otherwise}, \end{cases}$$

*where*

$$\beta = [f^{-1}(\alpha)(m+1)] - 1$$

*and $f \in \mathcal{F}_1 \cup \mathcal{F}_2$, namely $\mathbb{E}(\text{TPR}|\mathcal{T}^{cal})$ follows the beta distribution $\mathcal{B}e(m - \beta + 1, \beta)$.*

The proof of the Theorem 4.1 is presented in Appendix A.1. Theorem 4.1 indicates that the pre-specified level $\alpha$ and the size of the calibrated set $m$ play pivotal roles in determining the distributional characteristics of the g-BH algorithm. We visualize the probability distribution of $\mathbb{E}(\text{TPR}|\mathcal{T}^{cal})$ with different $\alpha$ and the size $m$ of calibrated set in Figure 2. From Figure 2, we find that if we choose a larger $\alpha$, the g-BH algorithm tends to achieve smaller TPR with appreciable probability. For example, with the calibration examples $m = 200$, we have $\mathbb{P}(\mathbb{E}(\text{TPR}|\mathcal{T}^{cal}) \leq 0.9) = 0.14$ for $\alpha = 0.08$ and $\mathbb{P}(\mathbb{E}(\text{TPR}|\mathcal{T}^{cal}) \leq 0.85) = 0.81$ for $\alpha = 0.12$. The reason behind this phenomenon is that a larger $\alpha$ induces the g-BH algorithm to adopt more aggressive decision-making strategies. In other words, the g-BH algorithm tends to classify more testing examples as OOD. More importantly, a small calibrated set causes the g-BH algorithm to achieve poor detection performance in terms of TPR. In contrast, a large calibrated set easily ensures a high TPR with a high probability. For example, with $\alpha = 0.05$, we obtain $\mathbb{P}(\mathbb{E}(\text{TPR}|\mathcal{T}^{cal}) \leq 0.9) = 0.32$ for $m = 200$, and $\mathbb{P}(\mathbb{E}(\text{TPR}|\mathcal{T}^{cal}) \leq 0.9) < 0.01$ for $m = 500$. Therefore, a large calibrated set significantly improves the performance of the g-BH algorithm. In Section 6, we also conduct extensive experiments on the real datasets to validate this conclusion.

# 5. Ensemble Generalized BH Algorithm for Small Calibrated Set

In Section 4, we demonstrate that a large calibrated set tends to improve the detection performance of the g-BH algorithm, but a small calibrated set easily leads to a poor results. In practice, we usually separate a portion of data from the training set to serve as the calibrated set. If the training set is small, a large calibrated set takes up more of the training data and further leads to the underfitting of neural networks. To address this problem, we propose the ensemble g-BH (eg-BH) algorithm to address this challenges faced by the g-BH algorithm.

Our motivation arises from the practical significance of p-values. A small calibrated set leads to under-representative empirical p-values, which fail to capture the distributional characteristics of the ID data. To address this issue, a natural approach is to generate multiple empirical p-values using the entire training set, thereby fully utilizing the available information. We then integrate these empirical p-values to make decisions.

For the given training data $\mathcal{T}$, we first partition $\mathcal{T}$ into $\mathcal{T}_1, \mathcal{T}_2, \cdots, \mathcal{T}_L$. Denote $\mathcal{T}_i^{cal} = \mathcal{T}_i$ and $\mathcal{T}_{-i}^{train} = \mathcal{T} \backslash \mathcal{T}_i$ where "$\backslash$" is the difference operation. Then, we train the score function $s(\cdot)$ based on $\mathcal{T}_{-i}^{train}$. For simplicity, we denote $s_i(\cdot)$ the score trained on $\mathcal{T}_{-i}^{train}$. Besides, denote by $|\mathcal{T}_i^{cal}|$ the size of $\mathcal{T}_i^{cal}$. Hence, for a testing example $X^{test}$, we enable to compute various empirical p-values using trained score function and calibrated set pairs $\{s_i(\cdot), \mathcal{T}_i^{cal}\}_{i=1}^L$:

$$\hat{p}_i(X^{test}) = \frac{|\{X^{cal} \in \mathcal{T}_i^{cal} : s_i(X^{cal}) \leq s_i(X^{test})\}| + 1}{|\mathcal{T}_i^{cal}| + 1}$$

for $i = 1, 2, \cdots, L$. After computing the empirical p-values $\hat{p}_1(X^{test}), \hat{p}_2(X^{test}), \cdots, \hat{p}_L(X^{test})$, the next problem is how to integrate these empirical p-values. A direct approach is to average them:

$$\bar{p}(X^{test}) = \frac{\hat{p}_1(X^{test}) + \hat{p}_2(X^{test}) + \cdots + \hat{p}_L(X^{test})}{L}.$$

Unfortunately, $\bar{p}(X^{test})$ does not necessarily satisfy the definition of the p-value (Rüschendorf, 1982; Meng, 1994).

To obtain a more general method of integrating the p-values, we first introduce a universal notion of average (Kolmogorov & Castelnuovo, 1930): given the p-values $\mathbf{p} = \{p_1, p_2, \cdots, p_L\}$ and weights $\mathbf{w} = \{w_1, w_2, \cdots, w_L\}$ where $w_i > 0$ and $\sum_{i=1}^L w_i = 1$, define

$$\Omega(\mathbf{p}, \mathbf{w}) = g^{-1}(w_1 g(p_1) + w_2 g(p_2) + \cdots, w_L g(p_L))$$

where $g(\cdot)$ is a continuous and strictly monotonic function, and $g^{-1}(\cdot)$ is its inverse function. If $w_i = \frac{1}{L}$, $\Omega(\cdot)$ is the

---

**Algorithm 1** eg-BH algorithm

1: **Input:** Training data $\mathcal{T}$, testing set $\mathcal{T}^{test} = \{X_1^{test}, X_2^{test}, \ldots, X_n^{test}\}$, prescribed level $\alpha \in (0, 1)$.
2: partition $\mathcal{T}$ into $\mathcal{T}_1, \mathcal{T}_2, \cdots, \mathcal{T}_L$, and let $\mathcal{T}_i^{cal} = \mathcal{T}_i$ and $\mathcal{T}_{-i}^{train} = \mathcal{T} \backslash \mathcal{T}_i$.
3: **for** $j = 1$ to L **do**
4:     Train the score function $s(x)$ on $\mathcal{T}_{-j}^{train}$, denote by $s_i(\cdot)$ the score trained on $\mathcal{T}_{-j}^{train}$.
5: **end for**
6: **for** $i = 1$ to $n$ **do**
7:     **for** $j = 1$ to $L$ **do**
8:         Compute the empirical p-values for testing example $X_i^{test}$ based on $s_j(\cdot)$ and $\mathcal{T}_j^{cal}$:

$$\hat{p}_{i,j} = \frac{|\{X \in \mathcal{T}_j^{cal} : s_j(X) \leq s_j(X_i^{test})\}| + 1}{|\mathcal{T}_j^{cal}| + 1}$$

9:     **end for**
10:     Integrate the empirical p-values $\hat{p}_{i,1}, \cdots, \hat{p}_{i,L}$ for $X_i^{test}$:

$$\bar{p}_i = \left( (\kappa + 1) \sum_{j=1}^L w_j \hat{p}_{i,j}^\kappa \right)^{\frac{1}{\kappa}} \qquad (7)$$

11: **end for**
12: Compute $i^* = \max\{i \in [n] : f(\bar{p}_{(i)}) \leq \frac{i}{n}\alpha\}$ where $\bar{p}_{(i)}$ is the $i$-th order statistic from the smallest to the largest for $\bar{p}_1, \cdot, \bar{p}_n$.
13: **Output:** Declare that $X_{(i)}^{test}$ is OOD if $i \leq i^*$, and the rests are ID.

---

arithmetic mean when $g(x) = x$; $\Omega(\cdot)$ is the geometric mean when $g(x) = \log x$; $\Omega(\cdot)$ is the harmonic mean when $g(x) = \frac{1}{x}$. For a random variable $X$, its $\alpha$-quantile is defined as

$$Q(X, \alpha) = \sup_{x \in \mathbb{R}} \{\mathbb{P}(X \leq x) < \alpha\}.$$

Clearly, $Q(X, 1)$ is the essential supremum of $X$. In addition, denote by $\mathcal{P}$ the set of all p-values. Suppose that the function $h(\cdot) : [0, 1]^L \to [0, \infty)$ is continuous and increasing, we define

$$Q^*(h, \mathbf{p}, \alpha) = \inf_{p_i \in \mathcal{P}} \{Q(h(p_1, \cdots, p_L), \alpha)\}$$

where $\mathbf{p} = \{p_1, \cdots, p_L\}$. The following theoretical results provide a concise method that integrates the multiple p-values.

**Theorem 5.1.** *Given the empirical p-values $p_1, p_2, \cdots .p_L$ and the function $g(x) = x^\kappa$ where $\kappa > 0$, then*

$$((\kappa + 1)(w_1 p_1^\kappa + w_2 p_2^\kappa + \cdots + w_L p_L^\kappa))^{\frac{1}{\kappa}}$$

*is a valid p-value. Specifically,*

$$\frac{2(p_1 + p_2 + \cdots + p_L)}{L}$$

*and*

$$\max\{p_1, p_2, \cdots, p_L\}$$

*are the valid p-values.*

The proof of Theorem 5.1 is presented in Appendix A.3. According to Theorem 5.1. we can choose appropriate function $g(x) = x^\kappa$ to integrate various empirical p-values for decision-making. The following theorem provide an consideration for the choice of $\kappa$ in $g(x)$.

**Theorem 5.2.** *Given the empirical p-values $p_1, p_2, \cdots . p_L$ and the function $g(x) = x^\kappa$, denote $w^* = \max\{w_1, w_2, \cdots, w_L\}$. If $w^* \leq \min\{\frac{1}{2}, \frac{1}{1+\kappa}, \frac{\kappa}{1+\kappa}\}$, then we have*

$$\sup_{p_i \in \mathcal{P}} \left\{ \mathbb{P}\left( \tilde{h}(\mathbf{p}) \leq \alpha \right) \right\} = \alpha.$$

*where $\tilde{h}(\mathbf{p}) = ((\kappa + 1)(w_1 p_1^\kappa + w_2 p_2^\kappa + \cdots + w_L p_L^\kappa))^{\frac{1}{\kappa}}$.*

The proof of Theorem 5.2 is presented in Appendix A.4. Theorem 5.2 indicates that if we choose $w^*$ such that $w^* \leq \min\{\frac{1}{2}, \frac{1}{1+\kappa}, \frac{\kappa}{1+\kappa}\}$,, the integrated p-value $\tilde{h}(\mathbf{p})$ can be exact, which benefits the improvement of power for the hypothesis testing algorithm. Based on the analysis above, we summarize our proposed method in Algorithm 1, called ensemble g-BH (eg-BH) algorithm.

# 6. Experiments

In this section, we aims to verify the effectiveness of Theorem 4.1 and the superiority of our proposed eg-BH algorithm over the g-BH algorithm. Our experimental framework is based on Ma et al. (2024) and use the same evaluation metrics. The experimental results show the superiority of the eg-BH algorithm over the g-BH algorithm on small calibrated set.

## 6.1. Experimental Settings

**Scores**. We choose two famous methods **MSP**(Hendrycks & Gimpel, 2017) and **Energy**(Liu et al., 2020) as the score functions in our method.

**Benchmarks**. We use CIFAR-10 (Krizhevsky et al., 2009) as ID data, and use CIFAR-100, ImageNet (Krizhevsky et al., 2017), SVHN (Netzer et al., 2011), Fashion-MNIST (F-MNIST) (Xiao et al., 2017), Places365 (Zhou et al., 2018) and MNIST (Deng, 2012), as OOD data.

**Metrics**. We use the same practical evaluation metrics as Ma et al. (2024), including TPR, FPR and F1-score.

**Model**. The score functions in this paper are based on the ResNet18 and WideResNet, respectively. We mainly

follow the experimental implementation in (Yang et al., 2022; Zhang et al., 2023a), and our codes are based on (Zhang et al., 2023a). More details are found in (Zhang et al., 2023a).

## 6.2. Impact of Calibrated Set on Generalized BH Algorithm

In this experiment, we aims to reveal how the calibrated set influences the detection performance of the g-BH algorithm. We first split the training data equally into two parts. One part is employed to train the neural networks for constructing the score function, and the other serves as the largest calibrated set $\mathcal{T}_M^{cal}$. Then, from $\mathcal{T}_M^{cal}$, we extract samples at various proportions $r$ to construct several relatively smaller calibrated sets, where $r = \{0.2, 0.3, \cdots, 1.0\}$. The experimental results of practical metrics based on the Energy (Liu et al., 2020) are presented in Tables 1 and 3. The results based on the MSP (Hendrycks & Gimpel, 2017) are presented in Tables 2 and 4. Because of the space limitation, all experimental results of using MNIST as OOD data are presented in Appendix B.

From Table 1, we find that with the increase of size for the calibrated set, the evaluation metrics TPR and F1-score considerably increases, accompanied by a marginal rise in FPR. For example, we use the SVHN as the OOD data, and use the Energy as our score function based on ResNet18. When sampling ratio $r = 0.2$, the F1-score, TPR and FPR of g-BH algorithm are $52.25\%$, $35.41\%$ and $0.05\%$, respectively. When $r = 0.8$, the corresponding F1-score, TPR and FPR are $75.78\%$, $61.51\%$ and $0.31\%$, respectively, which leads to the direct improvements of $23.53\%$ and $26.10\%$ for F1-score and TPR, at a negligible cost of $0.26\%$ increase in FPR. Notably, as shown in Tables 2, 3 and 4, this trend is consistent for other socre function MSP, network architecture WideResNet and OOD data. Therefore, large calibrated set improves the performance of the g-BH algorithm without the dependence on the distribution assumptions of OOD data. The above analysis demonstrates the effectiveness of Theorem 4.1.

## 6.3. Comparison between g-BH and eg-BH on Small Calibrated Set

In this experiment, we aim to compare the detection performance between vanilla g-BH algorithm and our proposed eg-BH algorithm on the small calibrated set. We first randomly divide the training data into $L$ equal parts. For the g-BH algorithm, one of these parts is used as the calibrated set. Note that a larger $L$ implies a smaller calibrated set. For our proposed method, we directly apply the strategies in the algorithm 1. When $L = 5$, the corresponding experimental results of practical metrics are presented in Tables 5 and 6 .

As tables 5 and 6 shown, we observe the FPR of our pro-

*Table 1.* Experimental results (%) of practical metrics on CIFAR-10 as ID data. **Energy** (Liu et al., 2020) is used as the score function based on the **ResNet18**. We compare the detection performance of g-BH algorithm with different sizes of calibrated set.

| Ratio | CIFAR-100 | | | TinyImageNet | | | SVHN | | | Place365 | | | F-MNIST | | |
|---|---|---|---|---|---|---|---|---|---|---|---|---|---|---|---|
| | F1 | TPR | FPR | F1 | TPR | FPR | F1 | TPR | FPR | F1 | TPR | FPR | F1 | TPR | FPR |
| 0.2 | 63.63 | 47.12 | 0.98 | 33.06 | 32.47 | 0.29 | 52.25 | 35.41 | 0.05 | 51.84 | 35.35 | 0.19 | 62.94 | 45.99 | 0.68 |
| 0.3 | 66.06 | 49.91 | 1.20 | 39.76 | 34.58 | 0.35 | 54.81 | 37.80 | 0.05 | 54.01 | 37.53 | 0.24 | 65.74 | 47.83 | 0.97 |
| 0.4 | 67.88 | 52.07 | 1.35 | 43.34 | 36.19 | 0.40 | 58.03 | 40.95 | 0.07 | 56.00 | 39.31 | 0.30 | 68.16 | 52.79 | 1.42 |
| 0.5 | 70.19 | 54.92 | 1.56 | 45.27 | 40.63 | 0.55 | 62.27 | 45.31 | 0.08 | 58.11 | 41.48 | 0.35 | 70.95 | 54.98 | 1.46 |
| 0.6 | 71.90 | 57.99 | 1.61 | 53.81 | 43.04 | 0.67 | 65.66 | 49.00 | 0.10 | 59.67 | 42.37 | 0.44 | 72.78 | 57.51 | 1.69 |
| 0.7 | 74.35 | 60.54 | 2.31 | 57.75 | 46.94 | 0.87 | 73.18 | 58.05 | 0.23 | 61.27 | 44.92 | 0.47 | 75.58 | 59.25 | 2.15 |
| 0.8 | 76.16 | 63.53 | 3.30 | 62.05 | 52.61 | 1.41 | 75.78 | 61.51 | 0.31 | 66.15 | 48.26 | 0.87 | 76.52 | 62.98 | 2.47 |
| 0.9 | 77.90 | 64.94 | 3.95 | 65.91 | 53.63 | 1.49 | 79.71 | 82.34 | 5.42 | 68.92 | 56.14 | 1.11 | 78.43 | 67.61 | 4.79 |
| 1 | 79.26 | 69.50 | 5.87 | 70.52 | 64.03 | 3.51 | 84.03 | 86.68 | 11.64 | 73.84 | 71.76 | 5.36 | 80.74 | 70.68 | 6.51 |

*Table 2.* Experimental results (%) of practical metrics on CIFAR-10 as ID data. **MSP** (Hendrycks & Gimpel, 2017) is used as the score function based on the **ResNet18**. We compare the detection performance of g-BH algorithm with different sizes of calibrated set.

| Ratio | CIFAR-100 | | | TinyImageNet | | | SVHN | | | Place365 | | | F-MNIST | | |
|---|---|---|---|---|---|---|---|---|---|---|---|---|---|---|---|
| | F1 | TPR | FPR | F1 | TPR | FPR | F1 | TPR | FPR | F1 | TPR | FPR | F1 | TPR | FPR |
| 0.2 | 62.27 | 45.62 | 0.91 | 48.49 | 32.47 | 0.29 | 51.55 | 34.76 | 0.04 | 50.31 | 33.81 | 0.16 | 62.12 | 45.36 | 0.68 |
| 0.3 | 64.89 | 48.56 | 1.11 | 50.08 | 33.95 | 0.33 | 55.67 | 38.63 | 0.06 | 53.42 | 36.75 | 0.23 | 64.89 | 48.45 | 0.89 |
| 0.4 | 67.88 | 52.07 | 1.35 | 52.38 | 36.19 | 0.40 | 58.03 | 40.95 | 0.07 | 54.19 | 37.51 | 0.25 | 67.84 | 51.93 | 1.17 |
| 0.5 | 70.19 | 54.92 | 1.56 | 53.81 | 37.64 | 0.45 | 61.22 | 44.21 | 0.08 | 57.02 | 40.37 | 0.33 | 70.20 | 54.83 | 1.39 |
| 0.6 | 71.81 | 57.00 | 1.76 | 55.64 | 39.53 | 0.51 | 65.66 | 49.00 | 0.10 | 60.16 | 43.68 | 0.42 | 71.81 | 56.89 | 1.56 |
| 0.7 | 74.35 | 60.54 | 2.31 | 58.74 | 42.97 | 0.67 | 67.76 | 51.39 | 0.12 | 62.77 | 46.63 | 0.53 | 74.37 | 60.50 | 2.20 |
| 0.8 | 75.26 | 62.03 | 2.82 | 62.05 | 46.94 | 0.87 | 71.67 | 56.10 | 0.17 | 65.41 | 49.92 | 0.74 | 75.19 | 61.76 | 2.51 |
| 0.9 | 78.49 | 67.87 | 5.06 | 65.42 | 51.84 | 1.33 | 75.78 | 61.51 | 0.31 | 68.82 | 54.55 | 1.09 | 76.98 | 64.80 | 3.56 |
| 1 | 81.28 | 77.63 | 13.40 | 69.63 | 70.05 | 6.23 | 79.71 | 86.34 | 11.64 | 74.26 | 70.23 | 5.18 | 79.58 | 70.25 | 6.30 |

*Table 3.* Experimental results (%) of practical metrics on CIFAR-10 as ID data. **Energy** (Liu et al., 2020) is used as the score function based on the **WideResNet**. We compare the detection performance of g-BH algorithm with different sizes of calibrated set.

| Ratio | CIFAR-100 | | | TinyImageNet | | | SVHN | | | Place365 | | | F-MNIST | | |
|---|---|---|---|---|---|---|---|---|---|---|---|---|---|---|---|
| | F1 | TPR | FPR | F1 | TPR | FPR | F1 | TPR | FPR | F1 | TPR | FPR | F1 | TPR | FPR |
| 0.2 | 63.52 | 50.19 | 4.53 | 46.42 | 35.22 | 2.49 | 38.08 | 36.97 | 1.97 | 48.78 | 36.41 | 1.32 | 64.30 | 47.85 | 1.77 |
| 0.3 | 65.84 | 50.88 | 5.99 | 48.92 | 36.49 | 3.24 | 52.95 | 39.86 | 2.49 | 52.64 | 37.58 | 1.79 | 65.89 | 51.21 | 2.25 |
| 0.4 | 67.22 | 52.79 | 7.24 | 50.15 | 38.84 | 4.08 | 56.71 | 41.48 | 3.05 | 54.88 | 39.73 | 2.04 | 67.19 | 53.79 | 2.86 |
| 0.5 | 68.85 | 56.17 | 8.06 | 50.79 | 40.16 | 4.99 | 57.46 | 43.15 | 3.41 | 56.49 | 40.89 | 2.65 | 70.53 | 55.84 | 3.59 |
| 0.6 | 70.28 | 59.06 | 8.55 | 53.84 | 42.55 | 5.71 | 59.25 | 45.89 | 3.97 | 58.34 | 44.25 | 3.02 | 73.48 | 59.39 | 4.01 |
| 0.7 | 73.43 | 61.72 | 9.09 | 54.75 | 44.63 | 6.38 | 62.94 | 47.74 | 4.33 | 61.75 | 48.84 | 3.48 | 75.59 | 62.41 | 4.64 |
| 0.8 | 74.58 | 64.66 | 13.27 | 56.88 | 48.18 | 7.45 | 64.77 | 50.91 | 4.82 | 64.48 | 53.79 | 3.99 | 77.11 | 64.29 | 5.29 |
| 0.9 | 77.53 | 67.18 | 14.62 | 60.13 | 66.09 | 10.04 | 65.96 | 52.76 | 5.12 | 66.52 | 56.68 | 4.51 | 79.49 | 69.49 | 6.89 |
| 1 | 78.97 | 78.73 | 22.79 | 61.52 | 67.75 | 12.84 | 67.18 | 56.57 | 5.79 | 70.16 | 71.41 | 8.79 | 83.09 | 82.23 | 16.52 |

*Table 4.* Experimental results (%) of practical metrics on CIFAR-10 as ID data. **MSP** (Hendrycks & Gimpel, 2017) is used as the score function based on the **WideResNet**. We compare the detection performance of g-BH algorithm with different sizes of calibrated set.

| Ratio | CIFAR-100 | | | TinyImageNet | | | SVHN | | | Place365 | | | F-MNIST | | |
|---|---|---|---|---|---|---|---|---|---|---|---|---|---|---|---|
| | F1 | TPR | FPR | F1 | TPR | FPR | F1 | TPR | FPR | F1 | TPR | FPR | F1 | TPR | FPR |
| 0.2 | 62.65 | 48.10 | 5.45 | 45.95 | 34.11 | 2.87 | 43.45 | 36.49 | 2.13 | 49.95 | 35.13 | 1.52 | 64.30 | 48.30 | 1.93 |
| 0.3 | 63.95 | 49.74 | 5.82 | 47.27 | 35.68 | 3.06 | 53.15 | 38.56 | 2.51 | 52.33 | 37.66 | 1.72 | 65.80 | 50.09 | 2.15 |
| 0.4 | 66.19 | 52.69 | 6.52 | 48.99 | 37.84 | 3.33 | 55.24 | 40.85 | 2.71 | 53.86 | 39.34 | 1.84 | 68.04 | 52.88 | 2.55 |
| 0.5 | 67.60 | 54.74 | 7.21 | 49.96 | 39.12 | 3.50 | 56.90 | 42.82 | 2.96 | 55.39 | 41.05 | 1.97 | 69.76 | 55.18 | 3.03 |
| 0.6 | 69.91 | 58.43 | 8.72 | 52.21 | 42.06 | 3.81 | 58.45 | 44.69 | 3.16 | 58.34 | 44.62 | 2.29 | 72.01 | 58.31 | 3.64 |
| 0.7 | 70.69 | 59.79 | 9.37 | 53.01 | 43.26 | 3.99 | 60.24 | 46.97 | 3.44 | 60.37 | 47.35 | 2.61 | 74.07 | 61.43 | 4.45 |
| 0.8 | 73.31 | 64.87 | 12.11 | 55.25 | 47.07 | 4.66 | 61.43 | 48.49 | 3.60 | 64.49 | 53.90 | 3.63 | 75.52 | 63.91 | 5.34 |
| 0.9 | 74.66 | 67.93 | 14.03 | 59.25 | 66.70 | 11.69 | 63.33 | 51.20 | 4.03 | 65.23 | 55.37 | 3.94 | 78.27 | 68.94 | 7.21 |
| 1 | 76.76 | 78.66 | 26.30 | 59.56 | 63.51 | 14.95 | 65.50 | 55.82 | 5.61 | 68.14 | 69.35 | 9.37 | 81.22 | 81.04 | 18.51 |

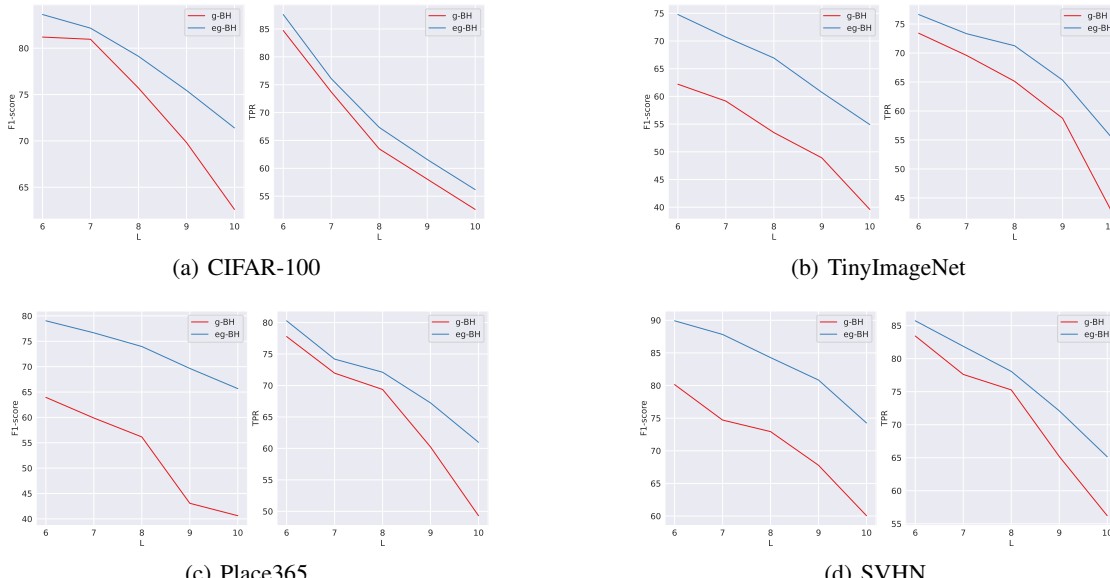

(a) CIFAR-100          (b) TinyImageNet

(c) Place365          (d) SVHN

*Figure 3.* Comparison between g-BH algorithm and our eg-BH algorithm in terms of F1-score and TPR. The x-axis corresponds to the number $L$ in Algorithm 1, and the y-axis represents the value of metrics.

*Table 5.* Experimental results (%) of practical metrics on CIFAR-10 as ID data. The score function is **Energy** based on ResNet18. We compare the performance between g-BH and eg-BH based on the same score function.

| Data | g-BH | | | eg-BH | | |
|---|---|---|---|---|---|---|
| | F1 | TPR | FPR | F1 | TPR | FPR |
| CIFAR-100 | 82.16 | 88.59 | 27.05 | 91.17 | 92.91 | 25.80 |
| TinyImageNet | 59.34 | 53.51 | 34.45 | 73.25 | 70.43 | 30.02 |
| SVHN | 75.16 | 59.13 | 29.06 | 89.00 | 79.43 | 31.30 |
| Place365 | 56.07 | 64.16 | 33.09 | 78.55 | 84.71 | 31.50 |
| F-MNIST | 84.89 | 87.91 | 23.82 | 88.10 | 90.26 | 24.05 |
| MNIST | 79.09 | 88.23 | 14.87 | 82.77 | 89.91 | 12.60 |
| Average | 69.45 | 73.59 | 27.06 | 83.81 | 84.61 | 25.88 |

*Table 6.* Experimental results (%) of practical metrics on CIFAR-10 as ID data. The score function is **MSP** based on ResNet18. We compare the performance between g-BH and eg-BH based on the same score function.

| Data | g-BH | | | eg-BH | | |
|---|---|---|---|---|---|---|
| | F1 | TPR | FPR | F1 | TPR | FPR |
| CIFAR-100 | 80.70 | 82.53 | 25.60 | 89.09 | 87.28 | 25.48 |
| TinyImageNet | 73.25 | 78.43 | 33.02 | 82.78 | 88.05 | 31.55 |
| SVHN | 86.00 | 87.43 | 4.30 | 91.17 | 92.15 | 4.06 |
| Place365 | 78.55 | 83.99 | 14.50 | 86.08 | 90.17 | 13.09 |
| F-MNIST | 85.26 | 82.63 | 21.79 | 90.11 | 89.27 | 19.05 |
| MNIST | 81.26 | 85.30 | 19.87 | 86.78 | 90.92 | 16.70 |
| average | 80.84 | 83.39 | 19.85 | 87.67 | 89.64 | 18.32 |

posed eg-BH algorithm achieves a certain degree of improvement compared with g-BH algorithm. More significantly, the TPR and F1-score are considerably improved. For example, when using Energy as score function and TinyImageNet as OOD data, compared to g-BH algorithm, our method reduce reduce the FPR from $34.45\%$ to $30.02\%$, improve the TPR by $16.92\%$ and the F1-score by $20.91\%$. Obviously, this improvement still exists for Different OOD data and score function MSP. The above analysis demonstrates the superiority of our method over the g-BH algorithm on small calibrated set.

To assess the impact of $L$ on both the g-BH algorithm and eg-BH algorithm. we set $L = \{6, 7, 8, 9, 10\}$ and conduct the corresponding experiments using Energy as score function based on the ResNet18. The experimental results are presented in Figure 3. From Figure 3, we find that our proposed method outperform the g-BH algorithm in terms of F1-score and TPR across different values of $L$. Moreover, for larger value $L$ (i.e. smaller calibrated set), the performance gap between our method and g-BH algorithm becomes more pronounced, especially when $L = 9$ and $L = 10$. This demonstrates the superiority of our proposed eg-BH algorithm over the g-BH algorithm on the smaller calibrated set.

## 7. conclusion

In this paper, we thoroughly analyze the g-BH algorithm and demonstrate that a large calibrated set improves the performance of the g-BH algorithm in terms of TPR, while a small calibrated set weakens its performance. To address this issue, we propose a novel eg-BH algorithm that integrates multiple p-values for decision-making. Extensive experiments demonstrate the validity of our theoretical results and

the superiority of our method over the g-BH algorithm.

## Acknowledgment

This work is supported by the Key R&D Program of Hubei Province under Grant 2024BAB038, the National Key R&D Program of China under Grant 2023YFC3604702,the Fundamental Research Funds for the Central Universities under Grant 2042025kf0045.

## Impact Statement

To our best knowledge, this work has no negative social impact. This work mainly provides a solid theoretical support for the field of the OOD detection and improves the performance of existing methods obviously. Hence, our work may promote the development of the related applications.

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

# A. Proofs

### A.1. Proof of Theorem 4.1

*Proof.* To obtain an explicit analytical solution, we assume that the testing data comes sequentially in a stream. Then, given a testing set $\mathcal{T}^{test} = \{X_1^{test}, X_2^{test}, \ldots, X_n^{test}\}$ consisting of ID data (TPR only focus on the detection performance of ID data), the expectation of TPR conditional on calibrated set $\mathcal{T}^{cal}$ for the g-BH algorithm can be expressed as

$$\mathbb{E}(\text{TPR}|\mathcal{T}^{cal}) = \mathbb{E}\left(\frac{1}{n}\sum_{i=1}^{n}\mathbb{1}(f(p(X_i^{test})) > \alpha)|\mathcal{T}^{cal}\right)$$

Note that the ID data $X_1^{test}, X_2^{test}, \ldots, X_n^{test}$ are independent and identically distributed, we have

$$\mathbb{E}(\text{TPR}|\mathcal{T}^{cal}) = \mathbb{E}\left(\mathbb{1}(f(p(X_1^{test})) > \alpha)|\mathcal{T}^{cal}\right)$$
$$= \mathbb{P}(f(p(X_1^{test})) > \alpha|\mathcal{T}^{cal}).$$

Note that the empirical p-value

$$\hat{p}(X_1^{test}) = \frac{\sum_{j=1}^{m}\mathbb{1}(s(X_j^{cal}) \leq s(X_i^{test})) + 1}{m+1}.$$

Since $f'(\cdot) \in \mathcal{F}_1 \cup \mathcal{F}_2$, $f'(\cdot) > 0$ and thus $f(\cdot)$ is increasing. Denote by $f^{-1}(\cdot)$ the inverse function of $f(\cdot)$. Then, we obtain

$$\mathbb{E}(\text{TPR}|\mathcal{T}^{cal}) = \mathbb{P}(f(p(X_1^{test})) > \alpha|\mathcal{T}^{cal})$$
$$= \mathbb{P}\left(\frac{\sum_{j=1}^{m}\mathbb{1}(s(X_j^{cal}) \leq s(X_i^{test})) + 1}{m+1} > f^{-1}(\alpha)|\mathcal{T}^{cal}\right)$$
$$= \mathbb{P}\left(\sum_{j=1}^{m}\mathbb{1}(s(X_j^{cal}) \leq s(X_i^{test})) > f^{-1}(\alpha)(m+1) - 1|\mathcal{T}^{cal}\right)$$
$$= \mathbb{P}\left(\frac{\sum_{j=1}^{m}\mathbb{1}(s(X_j^{cal}) \leq s(X_i^{test}))}{m} > \frac{f^{-1}(\alpha)(m+1) - 1}{m}|\mathcal{T}^{cal}\right)$$
$$= \mathbb{P}\left(\hat{F}(s(X_i^{test})) > \frac{f^{-1}(\alpha)(m+1) - 1}{m}|\mathcal{T}^{cal}\right)$$

Without loss of generality, we suppose that the random variable $s(X_1^{test})$ follows continuous distribution. Denote by $F(\cdot)$ the real cumulative distribution function of $s(X_1^{test})$ and by $\hat{F}^{-1}(\cdot)$ the inverse function of $\hat{F}(\cdot)$. In addition, we denote $[\cdot]$ the floor function. Then, we get

$$\mathbb{E}(\text{TPR}|\mathcal{T}^{cal}) = 1 - \mathbb{P}\left(\hat{F}(s(X_i^{test})) \leq \frac{f^{-1}(\alpha)(m+1) - 1}{m}|\mathcal{T}^{cal}\right)$$
$$= 1 - \mathbb{P}\left(\hat{F}(s(X_i^{test})) \leq \frac{[f^{-1}(\alpha)(m+1)] - 1}{m}|\mathcal{T}^{cal}\right)$$
$$= 1 - \mathbb{P}\left(s(X_i^{test}) \leq \hat{F}^{-1}(\frac{[f^{-1}(\alpha)(m+1)] - 1}{m})|\mathcal{T}^{cal}\right)$$
$$= 1 - F\left(\hat{F}^{-1}\left(\frac{[f^{-1}(\alpha)(m+1)] - 1}{m}\right)\right).$$

For simplicity, we denote $\beta = [f^{-1}(\alpha)(m+1)] - 1$. Note that $\hat{F}(X_{(\beta)}^{cal}) = \frac{\beta}{m}$. Therefore, we have

$$\mathbb{E}(\text{TPR}|\mathcal{T}^{cal}) = 1 - F(X_{(\beta)}^{cal}).$$

According to Eq. (5), $\mathbb{E}(\text{TPR}|\mathcal{T}^{cal}) = 1 - p(X_{(\beta)}^{cal})$.

To complete our proof, we need the following technical lemma.

**Lemma A.1.** *Suppose the continuous random variable $X$ have the cumulative distribution function $F(\cdot)$. Then, the random variable $F(X)$ follows the uniform distribution on $(0, 1)$.*

For the examples $X_1^{cal}, X_2^{cal}, \ldots, X_m^{cal}$, we denote $p(X_i^{cal}) = F(X_i^{cal})$, $i = 1, 2, \cdots, m$ and $p(X_{(\beta)}^{cal})$ is the $\beta$-th order statistic from the smallest to the largest. By Lemma A.1, $p(X_i^{cal})$ follows the uniform distribution on $(0, 1)$. Next, we aims to derive the probability density function of $p(X_{(\beta)}^{cal})$. According to the Definition of $p(X_i^{cal})$, $p(X_1^{cal}), p(X_2^{cal}), \cdots, p(X_m^{cal})$ are independent and identically distributed. Denote by $F_p(\cdot)$ the cumulative distribution function of $p(X_i^{cal})$. For any $x$ in the support set of $F_p(\cdot)$ and a sufficiently small $\delta$, we have

$$
\begin{aligned}
\mathbb{P}\left(x \le p(X_{(\beta)}^{cal}) < x + \delta\right) &= \mathbb{P}(\text{ one of the } p(X^{cal})'s \in [x, x+\delta) \text{ and } \beta - 1 \text{ of the others } < x) \\
&= \sum_{i=1}^{n} \mathbb{P}\left(p(X_i^{cal}) \in [x, x+\delta) \text{ and exactly } \beta - 1 \text{ of the others } < x\right) \\
&= n\mathbb{P}\left(p(X_1^{cal}) \in [x, x+\delta) \text{ and } \beta - 1 \text{ of the others } < x\right) \\
&= n\mathbb{P}\left(p(X_1^{cal}) \in [x, x+\delta)\right) \mathbb{P}(\beta - 1 \text{ of the others } < x) \\
&= n\mathbb{P}\left(p(X_1^{cal}) \in [x, x+\delta)\right)\left(\binom{m-1}{\beta-1}\mathbb{P}(p(X_1^{cal}) < x)^{\beta-1}P(p(X_1^{cal}) > x)^{m-\beta}\right)
\end{aligned}
\tag{8}
$$

Then, the probability density function of $p(X_{(\beta)}^{cal})$ is

$$
\begin{aligned}
f_\beta(x) &= \lim_{\delta \to 0} \frac{\mathbb{P}\left(x \le p(X_{(\beta)}^{cal}) < x + \delta\right)}{\delta} \\
&= m\binom{m-1}{\beta-1}F^{\beta-1}(x)(1 - F(x))^{m-\beta}F'(x) \\
&= \begin{cases} m\binom{m-1}{\beta-1}x^{\beta-1}(1-x)^{m-\beta} & \text{if } 0 < x < 1 \\ 0 & \text{otherwise.} \end{cases}
\end{aligned}
$$

Therefore, the probability density function of $\mathbb{E}(\mathrm{TPR}|\mathcal{T}^{cal}) = 1 - F(X_{(\beta)}^{cal})$ can be expressed as

$$
\begin{aligned}
f_{\mathbb{E}(\mathrm{TPR}|\mathcal{T}^{cal})}(x) &= f_\beta(1 - x) \\
&= \begin{cases} m\binom{m-1}{\beta-1}x^{m-\beta}(1-x)^{\beta-1} & \text{if } 0 < x < 1 \\ 0 & \text{otherwise.} \end{cases}
\end{aligned}
$$

The above result indicates that $\mathbb{E}(\mathrm{TPR}|\mathcal{T}^{cal})$ follows beta distribution with shape parameters $m - \beta + 1$ and $\beta$, which completes the proof.

$\square$

### A.2. Technical Lemmas and Their Proofs

**Lemma A.2.** *Suppose that the function $g(\cdot)$ is continuous and monotonically increasing on $[0, 1]$. Then, for any $\alpha \in (0, 1)$, we have*

$$
\int_0^\alpha g(x)\, dx \le \alpha \int_0^1 g(x)\, dx.
$$

*Proof.* Since function $g(\cdot)$ is increasing, $g(x)$ is integrable, namely, $\int_0^1 g(x)\, dx < \infty$. Note that

$$
\alpha \int_0^1 g(x)\, dx = \alpha \int_0^\alpha g(x)\, dx + \alpha \int_\alpha^1 g(x)\, dx
$$

then, we have

$$\int_0^\alpha g(x)\,dx - \alpha \int_0^1 g(x)\,dx = \int_0^\alpha g(x)\,dx - \alpha \int_0^\alpha g(x)\,dx - \alpha \int_\alpha^1 g(x)\,dx$$
$$= (1-\alpha)\int_0^\alpha g(x)\,dx - \alpha \int_\alpha^1 g(x)\,dx.$$

By the first mean value theorem for integration, there exist $\xi_1 \in (0,\alpha)$ and $\xi_2 \in (\alpha, 1)$ such that

$$\int_0^\alpha g(x)\,dx = \alpha g(\xi_1), \qquad \int_\alpha^1 g(x)\,dx = (1-\alpha)g(\xi_2).$$

Obviously, $\xi_1 \le \xi_2$. Since $g(x)$ is increasing, then $g(\xi_1) \le g(\xi_2)$. Therefore, we have

$$\int_0^\alpha g(x)\,dx - \alpha \int_0^1 g(x)\,dx = \alpha(1-\alpha)(g(\xi_1) - g(\xi_2)) \le 0,$$

namely, for any $\alpha \in (0,\ 1)$,

$$\int_0^\alpha g(x)\,dx \le \alpha \int_0^1 g(x)\,dx.$$

$\square$

Based on the lemma A.2, we have following lemma.

**Lemma A.3.** *Suppose that the function $g(\cdot)$ is continuous and monotonically increasing on $[0,\ 1]$. Then, for any $\alpha \in (0,\ 1)$, we have*

$$\mathbb{P}\left(\Omega(\mathbf{p}, \mathbf{w}) \le g^{-1}\left(\frac{1}{\alpha}\int_0^\alpha g(x)\,dx\right)\right) \le \alpha.$$

*Proof.* Denote $Y_i = g(p_i)$ where $p_i \in \mathcal{P}$. Without loss of generality, we assume that the p-values $p_1, \cdots, p_L$ are exact. Note that for any $t \in (0,\ 1)$, we have

$$\mathbb{P}\left(Y_i \le t\right) = \mathbb{P}\left(g(p_i) \le t\right) = \mathbb{P}\left(p_i \le g^{-1}(t)\right) = g^{-1}(t).$$

Therefore, $g^{-1}(\cdot)$ is the cumulative distribution function of $Y_i$. Based on the theoretical results in Bernard et al. (2014), we obtain

$$Q^*(h, \mathbf{p}, \alpha) = \inf_{p_i \in \mathcal{P}}\{Q(\Omega(\mathbf{p}, \mathbf{w}), 1)\}$$

where $h(p_1, \cdots, p_L) = \Omega(\mathbf{p}, \mathbf{w})$ and

$$\Omega(\mathbf{p}, \mathbf{w}) = g^{-1}\left(w_1 g(p_1) + w_2 g(p_2) + \cdots, w_L g(p_L)\right)$$

Note that $Q((w_1 g(p_1) + w_2 g(p_2) + \cdots, w_L g(p_L)), 1)$ is the essential supremum of $w_1 g(p_1) + w_2 g(p_2) + \cdots, w_L g(p_L)$, thus the following relations holds:

$$Q((w_1 g(p_1) + w_2 g(p_2) + \cdots, w_L g(p_L)), 1) \ge \mathbb{E}\left(w_1 g(p_1) + w_2 g(p_2) + \cdots, w_L g(p_L)\right).$$

Since $g(p_1), g(p_2), \cdots, g(p_L)$ are identically distributed sharing the cumulative distribution function $g^{-1}(\cdot)$, we have

$$\mathbb{E}\left(w_1 g(p_1) + w_2 g(p_2) + \cdots, w_L g(p_L)\right) = w_1 \mathbb{E}(g(p_1)) + w_2 \mathbb{E}(g(p_2)) + \cdots, w_L \mathbb{E}(g(p_L))$$
$$= (w_1 + w_2 + \cdots + w_L)\mathbb{E}(g(p_1))$$
$$= \mathbb{E}(g(p_1)) = \int_0^1 g(x)\,dx$$

By Lemma A.2, we get

$$Q((w_1g(p_1) + w_2g(p_2) + \cdots, w_Lg(p_L)), 1) \geq \frac{1}{\alpha} \int_0^\alpha g(x)\,dx$$

Because $g(\cdot)$ is continuous and increasing, $g^{-1}(\cdot)$ is also continuous and increasing. Hence, for any $p_i \in \mathcal{P}$ we have

$$Q(g^{-1}((w_1g(p_1) + w_2g(p_2) + \cdots, w_Lg(p_L))), 1) \geq g^{-1}\left(\frac{1}{\alpha}\int_0^\alpha g(x)\,dx\right),$$

namely, $g^{-1}\left(\frac{1}{\alpha}\int_0^\alpha g(x)\,dx\right)$ is the lower bound of $Q(g^{-1}((w_1g(p_1) + w_2g(p_2) + \cdots, w_Lg(p_L))), 1)$. Then, we get

$$Q^*(h, \mathbf{p}, \alpha) = \inf_{p_i \in \mathcal{P}} \left\{Q(g^{-1}((w_1g(p_1) + w_2g(p_2) + \cdots, w_Lg(p_L))), 1)\right\}$$

$$\geq g^{-1}\left(\frac{1}{\alpha}\int_0^\alpha g(x)\,dx\right).$$

According to the definition of $\alpha$-quantile, we obtain

$$\mathbb{P}\left(\Omega(\mathbf{p}, \mathbf{w}) \leq g^{-1}\left(\frac{1}{\alpha}\int_0^\alpha g(x)\,dx\right)\right) \leq \mathbb{P}\left(\Omega(\mathbf{p}, \mathbf{w}) \leq Q(\Omega(\mathbf{p}, \mathbf{w}), \alpha)\right) \leq \alpha.$$

$\square$

Lemma A.3 provide a significant region for the level $\alpha$. By Lemma A.3, we can choose appropriate function $g(\cdot)$ to integrate various p-values.

### A.3. Proof of Theorem 5.1

*Proof.* When $g(x) = x^\kappa$, $g^{-1}(x) = x^{\frac{1}{\kappa}}$. According to the Theorem A.3, we have

$$g^{-1}\left(\frac{1}{\alpha}\int_0^\alpha g(x)\,dx\right) = \left(\frac{1}{\kappa+1}\alpha^\kappa\right)^{\frac{1}{\kappa}} = (\kappa+1)^{-\frac{1}{\kappa}}\alpha.$$

and

$$\mathbb{P}\left(\Omega(\mathbf{p}, \mathbf{w}) \leq g^{-1}\left(\frac{1}{\alpha}\int_0^\alpha g(x)\,dx\right)\right)$$

$$= \mathbb{P}\left((w_1p_1^\kappa + w_2p_2^\kappa + \cdots + w_Lp_L^\kappa)^{\frac{1}{\kappa}} \leq (\kappa+1)^{-\frac{1}{\kappa}}\alpha\right)$$

$$= \mathbb{P}\left((\kappa+1)^{\frac{1}{\kappa}}(w_1p_1^\kappa + w_2p_2^\kappa + \cdots + w_Lp_L^\kappa)^{\frac{1}{\kappa}} \leq \alpha\right) \leq \alpha$$

Therefore,

$$(\kappa+1)^{\frac{1}{\kappa}}(w_1p_1^\kappa + w_2p_2^\kappa + \cdots + w_Lp_L^\kappa)^{\frac{1}{\kappa}}$$

is a valid p-value. Specifically, when $\kappa = 1$ and $w_i = \frac{1}{L}$,

$$(\kappa+1)^{\frac{1}{\kappa}}(w_1p_1^\kappa + w_2p_2^\kappa + \cdots + w_Lp_L^\kappa)^{\frac{1}{\kappa}} = \frac{2(p_1 + p_2 + \cdots + p_L)}{L}.$$

Denote $w_{k^*} = \max\{p_1, p_2, \cdots, p_L\}$, Note that

$$(\kappa+1)^{\frac{1}{\kappa}}(w_{k^*}p_{k^*}^\kappa)^{\frac{1}{\kappa}} \leq (\kappa+1)^{\frac{1}{\kappa}}(w_1p_1^\kappa + w_2p_2^\kappa + \cdots + w_Lp_L^\kappa)^{\frac{1}{\kappa}} \leq (\kappa+1)^{\frac{1}{\kappa}}(p_{k^*}^\kappa)^{\frac{1}{\kappa}}$$

further,

$$\lim_{\kappa \to \infty} (\kappa+1)^{\frac{1}{\kappa}}(w_{k^*}p_{k^*}^\kappa)^{\frac{1}{\kappa}} = \lim_{\kappa \to \infty} (\kappa+1)^{\frac{1}{\kappa}}(p_{k^*}^\kappa)^{\frac{1}{\kappa}} = p_{k^*}$$

Hence, when $\kappa \to \infty$, we have

$$\lim_{\kappa \to \infty} (\kappa+1)^{\frac{1}{\kappa}}(w_1p_1^\kappa + w_2p_2^\kappa + \cdots + w_Lp_L^\kappa)^{\frac{1}{\kappa}} = \max\{p_1, p_2, \cdots, p_L\}.$$

$\square$

### A.4. Proof of Theorem 5.2

*Proof.* According the proof of Theorem A.3, if $g(x) = x^\kappa$, we have

$$Q^*(h, \mathbf{p}, \alpha) = \inf_{p_i \in \mathcal{P}} \{Q(\Omega(\mathbf{p}, \mathbf{w}), 1)\} = \inf_{p_i \in \mathcal{P}} \left\{ Q((w_1 p_1^\kappa + w_2 p_2^\kappa + \cdots + w_L p_L^\kappa)^{\frac{1}{\kappa}}, 1) \right\}$$

Then,

$$(Q^*(h, \mathbf{p}, \alpha))^\kappa = \inf_{p_i \in \mathcal{P}} \{Q((w_1 p_1^\kappa + w_2 p_2^\kappa + \cdots + w_L p_L^\kappa), 1)\} \geq \frac{\alpha^\kappa}{\kappa + 1}.$$

Note that $\kappa > 0$, the probably density function of $g(p_i)$ is monotone on its support set. By Wang & Wang (2016),

$$(Q^*(h, \mathbf{p}, \alpha))^\kappa = \inf_{p_i \in \mathcal{P}} \{Q((w_1 p_1^\kappa + w_2 p_2^\kappa + \cdots + w_L p_L^\kappa), 1)\} = \frac{\alpha^\kappa}{\kappa + 1}.$$

if and only if the "mean condition"is satisfied:

$$w^* \alpha^\kappa \leq \frac{\alpha^\kappa}{\kappa + 1} \leq (w_1 + w_2 + \cdots + w_L)\alpha^\kappa - w^* \alpha^\kappa = (1 - w^*)\alpha^\kappa.$$

Equivalently, $w^* \leq \min\{\frac{1}{2}, \frac{1}{1+\kappa}, \frac{\kappa}{\kappa+1}\}$. Further, we have

$$Q^*(\tilde{h}, \mathbf{p}, \alpha) = \inf_{p_i \in \mathcal{P}} \left\{ Q((\kappa + 1)^{\frac{1}{\kappa}}(w_1 p_1^\kappa + w_2 p_2^\kappa + \cdots + w_L p_L^\kappa)^{\frac{1}{\kappa}}, 1) \right\} = \alpha, \tag{9}$$

where $\tilde{h}(\mathbf{p}) = (\kappa + 1)^{\frac{1}{\kappa}}(w_1 p_1^\kappa + w_2 p_2^\kappa + \cdots + w_L p_L^\kappa)^{\frac{1}{\kappa}}$. Next, based on the condition in Eq. (9), we aim to demonstrate

$$\sup_{p_i \in \mathcal{P}} \left\{ \mathbb{P}\left(\tilde{h}(\mathbf{p}) \leq \alpha\right) \right\} = \alpha.$$

If $Q^*(\tilde{h}, \mathbf{p}, \alpha) = \alpha$, for any $\alpha \in (0, 1)$ and arbitrary p-values $p_1, p_2, \cdots, p_L$ where $p_i \in \mathcal{P}$, we have $Q(\tilde{h}(\mathbf{p}), \alpha) \geq \alpha$ according to the definition of $Q^*(\tilde{h}, \mathbf{p}, \alpha)$. By the definition of $\alpha$-quantile, $\mathbb{P}(\tilde{h}(\mathbf{p}) < \alpha) \leq \alpha$. It follows that

$$\mathbb{P}\left(\tilde{h}(\mathbf{p}) \leq \alpha\right) \leq \mathbb{P}\left(\tilde{h}(\mathbf{p}) < \alpha + \delta\right) \leq \alpha + \delta,$$

Since $\delta$ is arbitrary, we have

$$\mathbb{P}\left(\tilde{h}(\mathbf{p}) \leq \alpha\right) \leq \alpha.$$

On the other hand, according to the definition of infimum, for any $\delta \in (0, 1)$, there exist the p-values $p_1^*, \cdots, p_L^* \in \mathcal{P}$ such that $\alpha \leq Q(\tilde{h}(\mathbf{p}^*), \alpha) < \alpha + \delta$, and thus $\mathbb{P}(\tilde{h}(\mathbf{p}^*) \leq \alpha + \delta) \geq \alpha$ where $\mathbf{p}^* = \{p_1^*, \cdots, p_L^*\}$. Since $\delta$ is arbitrary, then we have

$$\sup_{p_i \in \mathcal{P}} \left\{ \mathbb{P}\left(\tilde{h}(\mathbf{p}) \leq \alpha\right) \right\} = \alpha.$$

$\square$

## B. Additional Experimental Results

In this section, we present additional experimental results. The results on MNIST as OOD data are presented in Tables 7. Table 7 shows the same conclusions as those of tables in main text.

*Table 7.* Experimental results (%) of practical metrics on CIFAR-10 as ID data. The MNIST is OOD data. **Energy** and **MSP** are used as the score functions. We compare the detection performance of g-BH algorithm with different sizes of calibrated set.

| Score | Energy | | | | | | MSP | | | | | |
|---|---|---|---|---|---|---|---|---|---|---|---|---|
| | ResNet18 | | | WideResNet | | | ResNet18 | | | WideResNet | | |
| Ratio | F1 | TPR | FPR | F1 | TPR | FPR | F1 | TPR | FPR | F1 | TPR | FPR |
| 0.2 | 63.48 | 47.94 | 4.94 | 63.16 | 45.87 | 1.98 | 62.87 | 48.25 | 5.25 | 62.46 | 46.45 | 2.29 |
| 0.3 | 65.15 | 50.15 | 5.78 | 65.09 | 49.12 | 2.35 | 64.23 | 50.10 | 5.90 | 64.95 | 49.40 | 2.72 |
| 0.4 | 67.57 | 52.95 | 6.95 | 66.97 | 53.99 | 3.04 | 66.67 | 53.69 | 6.37 | 67.95 | 53.13 | 3.25 |
| 0.5 | 69.53 | 57.39 | 7.27 | 70.15 | 56.75 | 3.59 | 66.67 | 54.68 | 7.59 | 69.76 | 55.61 | 3.82 |
| 0.6 | 72..94 | 59.82 | 8.54 | 71.59 | 59.44 | 4.25 | 69.91 | 58.59 | 9.02 | 70.88 | 57.20 | 4.19 |
| 0.7 | 73.82 | 62.29 | 10.11 | 75.53 | 63.17 | 6.11 | 71.19 | 60.79 | 10.00 | 72.91 | 60.20 | 4.94 |
| 0.8 | 74.75 | 65.87 | 13.68 | 77.46 | 66.57 | 6.79 | 71.82 | 62.19 | 11.00 | 74.83 | 63.37 | 6.00 |
| 0.9 | 75.59 | 68.57 | 15.35 | 79.85 | 69.44 | 8.26 | 73.52 | 66.55 | 14.49 | 78.13 | 69.69 | 8.70 |
| 1 | 76.81 | 75.49 | 19.54 | 81.74 | 73.39 | 10.79 | 75.59 | 81.51 | 34.14 | 79.15 | 72.47 | 10.65 |

