# OpenReview forum: "A Closer Look at Generalized BH Algorithm for Out-of-Distribution Detection"
_ICML.cc/2025/Conference — ICML 2025 poster_

### Official Review · Reviewer_rS6P · 2025-03-12

**Overall Recommendation:** 2

**Summary:**

The paper investigates methods for setting a decision threshold for out-of-distribution (OOD) detection using a calibration set, ensuring that the resulting detector achieves a desired performance on unseen data. The authors examine the g-BH algorithm, a recently proposed method for threshold selection [Ma 2024], and extend it in two ways. First, they characterize the distribution of the expected True Positive Rate for decision rules using thresholds set by the g-BH algorithm, demonstrating that the TPR follows a Beta distribution with parameters dependent on the calibration set size and the target $\alpha$-level. Second, they propose an ensemble-based algorithm to address the inefficiency of the g-BH algorithm when using small calibration sets. Both contributions are empirically validated on standard OOD benchmarks.

## update after rebuttal
I acknowledge that I failed to fully grasp the setup and purpose of the proposed algorithm based on the current presentation. While I did not consult [Ma et al., 2024], which may provide additional context, I believe the paper should be self-contained and provide a clear and complete definition of the problem it addresses.

The authors' rebuttal offered some clarification, but key aspects remain unclear to me. For instance, the claim that "a larger calibration set improves the performance of the g-BH algorithm in terms of TPR" is difficult to interpret meaningfully without addressing other performance metrics such as FPR. Since the method revolves around setting a decision threshold on a small calibration set, reporting improvements in TPR alone seems insufficient for a full evaluation.

The paper may contain valuable ideas, but the current presentation does not clearly convey them to me. This limits my ability to fully understand the core contributions and assess their significance. While I recognize the possibility that the misunderstanding lies on my side, I made an effort to engage with the paper.

**Claims And Evidence:**

A central claim, reiterated multiple times in the paper, is that larger calibration sets enhance the performance of the g-BH algorithm in terms of the true positive rate (TPR). However, this claim is somewhat misleading and, in my opinion, fundamentally incorrect, as it does not align with the intended purpose of the g-BH algorithm or its guarantees. The algorithm is designed solely to set the decision threshold such that the actual false discovery rate (FDR) or TPR does not exceed a specified threshold at a given confidence level.

**Essential References Not Discussed:**

No.

**Experimental Designs Or Analyses:**

As previously noted, the design and analysis of the experiments have the flaws mentioned earlier.

**Methods And Evaluation Criteria:**

The authors devote a significant portion of the paper to empirically analyzing how the calibration set size affects the performance of the OOD detector configured by the g-BH algorithm. However, the experimental setup is not clearly described, and its purpose remains somewhat unclear. In particular, the confidence level (p-value), and the desired FDR threshold are not consistently defined across all experiments.

Furthermore, since the core issue addressed in the paper is the randomness introduced by small calibration sets, the experiments should account for this variability. This could be done by running multiple trials and reporting summary statistics (such as the mean and standard deviation) rather than presenting single-value metrics (e.g., FPT, F1), which are merely realizations of random variables with high variance.

**Other Comments Or Suggestions:**

No.

**Other Strengths And Weaknesses:**

As mentioned earlier, the problem to be addressed is not clearly defined, the claims lack clarity and precision, and the descriptions of the algorithms and experiments are insufficiently detailed. Examples of issues related to problem formulation and vague claims have already been provided. Additionally, several key details are missing regarding the algorithms and experimental setup. For instance, it is unclear how the weights of the ensemble algorithm are determined, what $n_0$ represents in Theorem 5.5 and Corollary 5.6, and how the p-values and FDR thresholds were set across all experiments.

Characterizing the distribution of the true positive rate (TPR) for the g-BH algorithm, along with the analysis of the proposed ensemble algorithm, could provide solid contributions. However, the paper's presentation is far from ideal, making it difficult to thoroughly assess its validity and clearly grasp its main message.

**Questions For Authors:**

One of the shortcomings of the paper is the lack of a clear problem definition, which should be stated explicitly at the beginning. Based on my understanding, the goal is to set a threshold on the calibration set such that the False Discovery Rate (the proportion of true OOD samples incorrectly identified as OOD by the detector) on unseen data remains below a specified threshold with at least $1-\alpha$ confidence. Can you confirm if this is the intended problem being addressed?

**Relation To Broader Scientific Literature:**

The authors cite relevant literature. I see a connection between Theorem 4.1 and a similar problem in conformal prediction, as discussed earlier. However, I do not view the omission of this reference as a significant issue.

**Theoretical Claims:**

I did not verify the proofs in detail, but I believe Theorem 4.1 is correct. The setup and results closely resemble those found in problems related to characterizing the coverage of a conformal predictor. Specifically, the claim that coverage—computed in the same manner as in the BH algorithm—follows a Beta distribution has been previously established in:
Vovk, "Conditional validity of inductive conformal predictor," ACML, 2012.

---

> ### Author Rebuttal · Authors · 2025-03-30
>
> **Q1**: about the claim and goal of our paper.
>
> **Ans1**: We first emphasize that FDR is closely related to TPR and FPR. Based on [1], we have
> $$ FDR = E(\frac{1}{1 + \frac{P}{N}\cdot \frac{1-FPR}{1-TPR} }) $$
> where P is the number of ID data in test set and N is the number of OOD data in test set. Obviously, larger TPR and smaller FPR leads to smaller FDR. [1]  points out that traditional decision rule just considers TPR when choosing decision threshold. By contrast, FDR can balance the TPR and FPR. __By controlling FDR, g-BH can better trade off the performance between ID and OOD data and further improve total decision performance (F1)__ (The details can be seen in third paragraph of Section 5.2 of [1]). The experiments in [1] also show that they care more about TPR, FPR and F1, instead of only controlling FDR.  The meaning of our claim is that larger calibrated set can improve the TPR of g-BH and decrease its FPR, leading to smaller FDR and larger F1. Our Theorem 4.1 and experimental results of Figure 1 and Tables 1-4 support this claim. Hence, our claim is align with the purpose of g-BH, and is correct without misleading.
>
> The goal of our paper is to __deeply study the influence of calibrated set on g-BH__, which has been repeatedly pointed out in Abstract (line 20-24) and Introduction (line 32-38, right). Overall, our paper consists two parts: 1, we find that small calibrated set degrades the performance of g-BH, and establish Theorem 4.1 and conduct extensive experiments (Tables 1-4) to verify this phenomenon. 2, we propose the eg-BH algorithm  to solve the issue caused by small calibrated set. The performance of hypothesis testing algorithm  depends on precise p-value. Since the distribution of ID data is unknown, [1] directly uses empirical p-value to estimate real p-value.  Glivenko-Cantelli theorem indicates that this estimation method performs well for large calibrated set. When calibrated set is small, the estimation error of p-value increases, leading to a poor performance of g-BH. Our core goal is to mitigate estimation error problem of p-value caused by small calibrated set. The entire Section 5 focuses on discussing how to obtain a good computation method of p-value based on small calibrated set. Combining our estimation method of p-value and g-BH, this algorithm is called eg-BH. Hence, our core goal is not simply to address the randomness caused by calibrated set, nor to set a threshold such FDR remains below a specified threshold.
>
> **Q2**: about the understanding of some concepts.
>
> __Ans2__: we first give a correct interpretation of FDR. FDR is the expectation of false discovery proportion (FDP). FDP can be expressed as A/B where A is the number of ID examples in test set falsely classified as OOD and B is the number of examples classified as OOD in test set. Hence,
> "False Discovery Rate (the proportion of true OOD samples incorrectly identified as OOD by the detector) on unseen data" is incorrect.  Besides, FDR is a real number without randomness. "at a given confidence level" is often used to describe random event. So, "at a given confidence level" is not suitable to describe FDR control. Different from conformal prediction, in hypothesis testing, p-value can not be called "confidence level". $\alpha$ is only upper bound of FDR  and not used to describe confidence level. Moreover, p-value is the function of socre of testing example, instead of hyper-parameter, and thus can not be prespecified.
>
> **Q3**: about experimental settings and some notations.
>
> **Ans3** Our code is based on [2] and we use same experimental setup as [2],  which has been emphasized in Section 6.1 (line 362-363, left). The details, such as optimizer, learning rate, epoch, can be found in
> Section 4.1 of [2]. Besides,  the code of [2] run 5 times for every model, and all experimental results are the mean of 5 trials. You can find these information in Section 4.2 of [2]. We will provide these details in Appendix.
> Our theoretical framework is based on [1] and we use the same notations and definitions as [1], which have been emphasized in Section 3 (line 102-104, right). The computation method of empirical p-value has been clearly presented in Section 5 (line 196-198, right) and Algorithm 1 (step 8) for two times. Following [1], $\alpha$ is set as 0.05 (see Section 2.3 of [1]). The weights in eg-BH is 1/L, which has been clearly pointed out in Theorem 5.3. We will describe the weights in Experiments again. $n _0$ is the number of ID data in test set, which has clear description in the proof of Theorem 5.5. You can also find its detailed description in Sections 2.2 and 3 of [1].
>
>
> [1] A Provable Decision Rule for Out-of-Distribution Detection
>
> [2] OpenOOD: Benchmarking Generalized Out-of-Distribution Detection

---

### Official Review · Reviewer_5FFL · 2025-03-13

**Overall Recommendation:** 4

**Summary:**

The paper explores the role of the calibrated set in the performance of the g-BH algorithm for OOD detection. Theoretical results indicates the large calibrated set will improve the performance of the g-BH algorithm but small calibrated set tends to degrade the performance of
It. Then, the authors propose a novel eg-BH algorithm to tackle the limitations of g-BH algorithm on the small calibrated set. Finally, the authors conduct extensive experiments to demonstrate the correctness of the theoretical results and the validity of proposed method. In summary, this paper represents a meaningful step forward in OOD detection study and provides valuable insights into multiple hypothesis testing applications.

**Claims And Evidence:**

Yes. This paper first verify the new finding about the g-BH algorithm on the small calibrated set by rigorous theoretical analysis and numerous experiments on real-world datasets. Then, this paper conducts extensive experiments to demonstrate the superiority of the proposed method over the g-BH algorithm on the small calibrated set.

**Essential References Not Discussed:**

No, the paper includes all essential and relevant references.

**Experimental Designs Or Analyses:**

Yes. In experiments, this paper first verifies the correctness of theoretical results. The experimental results show the same conclusions as Theorem 4.1. Then, this paper conducts extensive experiments to demonstrate the superiority of the proposed method over the g-BH algorithm on the small calibrated set.

**Methods And Evaluation Criteria:**

Yes. This paper uses many evaluation criteria, including TPR, FPR, F1-score, AUROC, AUPR and FPR95. These criteria are suitable for the OOD detection problem.

**Other Comments Or Suggestions:**

No

**Other Strengths And Weaknesses:**

Strengths

- This paper reveals the influence of calibrated set on the g-BH algorithm.

- This paper establishes a mathematical relationship between the size of the calibrated set and the conditional TPR expectation of the g-BH algorithm.

- This paper proposes a novel method to improve OOD detection performance on the small calibrated set by aggregating multiple empirical p-values.

- This paper show the impact of different calibrated set sizes using various OOD datasets (CIFAR-10, SVHN, TinyImageNet, etc.). Besides, this paper shows that the proposed method consistently outperforms g-BH algorithm on the small calibrated set.


Weaknesses

I have not found any obvious weaknesses. It would be more comprehensive if the following questions are addressed:

- The paper uses numerous terms from hypothesis testing, such as type-1 error, significant level and FDR, which may lead to difficulties for the readers unfamiliar with hypothesis testing. So the author should provide more explanations for these concepts.

- In Theorem 4.1, what are the assumptions about the score functions?

- In Algorithm 1, how do you choose the number L?

**Questions For Authors:**

See above

**Relation To Broader Scientific Literature:**

This paper finds the limitations of the g-BH algorithm on the small calibrated set and proposes a novel method to address this problem.

**Theoretical Claims:**

Yes. I check the proofs in the Appendix. especially for Theorem 4.1 and Theorem 5.3, since these theorems are the key contributions of this paper.

- Theorem 4.1 derives the distribution of conditional TPR on calibrated set, the distribution parameters are determined by the significance level and size of the calibrated set;

- Theorem 5.3 provides a concrete method for integrating multiple p-values, which is the basis of the proposed eg-BH algorithm.

---

> ### Author Rebuttal · Authors · 2025-03-30
>
> __Weakness 1__: about the interpretations of some concepts, including type-1 error, significant level and FDR.
>
> __Ans1__: we interpret these concepts as follows:
>
> Significance level: If the probability of obtaining a result as extreme as the one obtained, supposing that the null hypothesis were true, is lower than a pre-specified cut-off probability (for example, 5%), then the result is said to be statistically significant (the null hypothesis is rejected), and the cut-off probability is called significant level.
>
> Type-1 error and FDR: In statistical hypothesis testing, a type-1 error is the rejection of the null hypothesis when it is actually true.
> FDR can be considered as the generalization of the probability of type-1 error in single hypothesis testing. In multiple testing, the null hypotheses rejected by the detection algorithms are called discoveries. FDR is used to describe the expected proportion of erroneous discoveries among all discoveries.
>
> __Weakness 2__: in Theorem 4.1, what are the assumptions about the score functions?
>
> __Ans2__. We just require that the score function is continuous, without more strong assumptions. For example, two famous score functions MSP and Energy satisfy this demand.
>
> __Weakness 3__: in Algorithm 1, how do you choose the number L?
>
> __Ans3__: In practice, we can obtain a well performance of OOD detection when L is set to 3 or 4.

---

### Official Review · Reviewer_dWQD · 2025-03-13

**Overall Recommendation:** 4

**Summary:**

Based on the recent work [1], this paper aims to study the influence of the calibrated set on the generalized BH (g-BH) algorithm for out-of-distribution (OOD) detection task. By theoretical analysis and experimental results on the real data, the authors show that the small calibrated set tends to degrade the performance of the g-BH algorithm. Then, the authors propose an enhanced approach, the ensemble g-BH (eg-BH) algorithm, which integrates multiple empirical p-values to solve this issue. The claims in this paper are built on strong theoretical foundations and supported by extensive empirical validation

**Claims And Evidence:**

Yes.

**Essential References Not Discussed:**

No, the paper includes all essential and relevant references.

**Experimental Designs Or Analyses:**

Yes. they seems quite sound.

**Methods And Evaluation Criteria:**

Yes.

**Other Comments Or Suggestions:**

No

**Other Strengths And Weaknesses:**

Strengths

(1) Theoretical analysis. The authors develop a novel theoretical understanding of the role of calibrated set in the g-BH algorithm.

(2) Novel method. The proposed method extends g-BH algorithm by integrating multiple empirical p-values, mitigating the problem due to small calibrated set.

(3) Extensive performance. The experimental setup, including the variation of calibrated set sizes, provides strong empirical support for the theoretical results. Additionally, the the authors use different score functions, such as Energy-based and Maximum Softmax Probability, to assess the robustness of the proposed approach on various benchmarks.. The evaluation is comprehensive, ensuring the generalizability of the results.


Weaknesses

(1) The proposed method requires a hold-out set. However, since the hold-out set consists of ID examples, this is not a significant restriction.

(2) Why does the authors only use the empirical p-values in Algorithm 1?

(3) Suggest checking and standardizing the format in Section 4.

**Questions For Authors:**

See Weaknesses

**Relation To Broader Scientific Literature:**

This paper provides a new decision framework based on the existing score functions, which enables to be adapted to small calibrate set.

**Theoretical Claims:**

Yes. I check some proofs in Appendix, but do not carefully read them step by step.

---

> ### Author Rebuttal · Authors · 2025-03-30
>
> __Weakness 1__: the proposed method requires a hold-out set. However, since the hold-out set consists of ID examples, this is not a significant restriction.
>
> __Ans1__: We emphasize that the calibrated set consists of ID data, without the need of OOD data.  We directly extract some examples from training data to construct calibrate set. Hence, it is easy to obtain a calibrated set for our proposed method.
>
> __Weakness 2__: why does the authors only use the empirical p-values in Algorithm 1?
>
> __Ans2__: We must clarify that in practice, any valid computation methods that satisfy the definition of p-values can be used. Since the distribution of ID data is unknown, we choose a non-parametric method: empirical p-values to estimate real p-values. Essentially, it is equivalent to use empirical distribution to estimate real distribution.
>
>
> __Weakness 3__: Suggest checking and standardizing the format in Section 4.
>
> __Ans3__: Thanks for your careful review. We have checked the typos in Section 4. We will fix these typos in new version.

---

> > ### Comment · Reviewer_dWQD · 2025-04-02
> >
> > Thanks author for the response. They have answered my questions well. I keep my positive score.

---

### Official Review · Reviewer_ZPgt · 2025-03-13

**Overall Recommendation:** 5

**Summary:**

This paper investigates the impact of the calibrated set on the generalized BH (g-BH) algorithm[1] for Out-of-Distribution (OOD) detection. The authors provide a theoretical analysis showing that the conditional expectation of the true positive rate (TPR) follows a beta distribution, demonstrating that a small calibrated set negatively affects performance of g-BH algorithm. To address this problem, they introduce the ensemble g-BH (eg-BH) algorithm, which integrates multiple empirical p-values for decision-making. Extensive experiments validate the theoretical findings, and show that eg-BH algorithm outperforms g-BH algorithm, particularly on small calibrated set.

**Claims And Evidence:**

Yes.the claims made in the submission appear to be supported by clear and convincing evidence.

**Essential References Not Discussed:**

No.

**Experimental Designs Or Analyses:**

Yes. The experimental designs and analyses are sound.

**Methods And Evaluation Criteria:**

Yes. the proposed method makes sense for the OOD detection.

**Other Comments Or Suggestions:**

No.

**Other Strengths And Weaknesses:**

The analytical framework of this paper is well-grounded in the statistical hypothesis testing. Hence, the conclusions have strong theoretical guarantee. The paper presents a rigorous theoretical analysis for the g-BH algorithm, demonstrating that small calibrated set tends to weaken the performance of it. The proposed eg-BH algorithm effectively enhances OOD detection performance on small calibrated set by integrating multiple empirical p-values, compared with g-BH algorithm. Extensive experimental results demonstrate the effectiveness of the proposed method.

Some weaknesses are listed below:

W1: The proposed eg-BH algorithm depends on multiple p-values. the advantages of integrating multiple p-values should be discussed.

W2 The proofs in line 572 - line 583 in appendix should be more detailed.

**Questions For Authors:**

What are the advantages of controlling FDR?

**Relation To Broader Scientific Literature:**

This paper focus on studying the influence of calibrated set on the performance of the g-BH algorithm. Besides, to address the problem caused by small calibrated set, this paper proposes the eg-BH algorithm to integrate the multiple empirical p-values for making decision.

**Theoretical Claims:**

Yes. I check some proofs, including Theorem 4.1, Lemma 5.1, Theorem 5.2 and Theorem 5.3.

---

> ### Author Rebuttal · Authors · 2025-03-30
>
> __W1__: The proposed eg-BH algorithm depends on multiple p-values. the advantages of integrating multiple p-values should be discussed
>
> __Ans1__.   A small calibrated set leads to under-representative empirical p-values, which fail to capture the distributional
> characteristics of the ID data. To address this issue, we utilize the available
> information in training set to generate multiple empirical p-values. We then integrate these empirical p-values to a single p-value with more  information of ID data, making it more discriminative.
>
> __W2__. The proofs in line 572 - line 583 in appendix should be more detailed.
>
> __Ans2__. Following your suggestions, the detailed steps are as follows:
>
> $$
>  \begin{aligned}
>     \mathbb{E}(\mathrm{TPR}| \mathcal{T}^{cal}) & =  \mathbb{P}(f(p(X_{1}^{test})) > \alpha  | \mathcal{T}^{cal}) \\\\
>     & = \mathbb{P}\left( \frac{ \sum_{j=1}^m \mathbf{1}(s(X_{j }^{cal})\leq  s(X_{i}^{test}))+1}{m+1} > f^{-1} (\alpha)  | \mathcal{T}^{cal} \right) \\\\
>     & = \mathbb{P}\left(  \sum_{j=1}^m \mathbf{1}(s(X_{j }^{cal})\leq  s(X_{i}^{test})) > f^{-1} (\alpha)(m+1 ) - 1  | \mathcal{T}^{cal} \right) \\\\
>     & = \mathbb{P}\left(  \frac{ \sum_{j=1}^m \mathbf{1}(s(X_{j }^{cal})\leq  s(X_{i}^{test}))}{m} > \frac{f^{-1} (\alpha)(m+1 ) - 1 }{m} | \mathcal{T}^{cal} \right) \\\\
>     & =  \mathbb{P}\left( \hat{F} ( s(X_{i}^{test}))  > \frac{f^{-1} (\alpha)(m+1 ) - 1 }{m}   |\mathcal{T}^{cal} \right)
>   \end{aligned}
>   $$
>
> __Q1__. What are the advantages of controlling FDR?
>
>   __Ans3__. It is well-known that there is a tradeoff between the detection performance of ID and OOD examples for a trained score functions. Therefore, we cannot only consider the true positive rate (TPR) or false positive
> rate (TPR) when designing the OOD detection algorithm. Factually, an ideal OOD detection algorithm should achieve low FPR while maintaining a high TPR, which leads to a small FDP. Thus, controlling FDR can achieve a well tradeoff between the detection performance of both ID and OOD examples.

---

> > ### Comment · Reviewer_ZPgt · 2025-04-02
> >
> > My questions have been addressed. Thanks for the reply.

---

### Decision · Program_Chairs · 2025-05-01

**Decision:**

Accept (poster)

**Comment:**

The paper analyzes the relationship between the size of the calibration set and the expectation of true positive rate (TPR) when performing out-of-distribution detection based on empirical p-value computed over the calibration set. To address the limitation caused by small calibration sets, the authors propose a novel detection algorithm that utilizes multiple empirical p-values computed over multiple calibration sets, which are obtained by partitioning the data. The proposed method is guaranteed to control the false discovery rate (FDR) at a prescribed level. Theoretical claims are supported by extensive experiments.

Overall, reviewers agree that the paper makes a meaningful contribution to the development of OOD detection theory, and that the proposed method is supported both theoretically and empirically. Meanwhile, authors are encouraged to enhance clarity in the presentation, especially provide more explanation on the statistical concepts from hypothesis testing.